# The Effect of Crystallinity on the Toughness of Cast Polyamide 6 Rods with Different Diameters

**DOI:** 10.3390/polym12020293

**Published:** 2020-02-02

**Authors:** Miklós Odrobina, Tamás Deák, László Székely, Tamás Mankovits, Róbert Zsolt Keresztes, Gábor Kalácska

**Affiliations:** 1Institute for Mechanical Engineering Technology, Szent István University, Páter Károly u. 1., H-2100 Gödöllő, Hungary; keresztes.robert@gek.szie.hu (R.Z.K.); kalacska.gabor@gek.szie.hu (G.K.); 2R&D Engineer, SimpaTec GmbH; Aspenhaustr. 5, D-72770 Reutlingen, Germany; t.deak@simpatec.com; 3Department of Mathematics, Institute of Mathematics and Basic Science, Szent István University, Páter Károly u. 1., H-2100 Gödöllő, Hungary; szekely.laszlo@gek.szie.hu; 4Department of Mechanical Engineering, Faculty of Engineering, University of Debrecen, Ótemető u. 2-4., H-4028 Debrecen, Hungary; tamas.mankovits@eng.unideb.hu

**Keywords:** degree of crystallinity, magnesium-catalyzed, polyamide 6, Charpy’s impact test, toughness

## Abstract

The present paper concentrates on the toughness and the degree of crystallinity of the magnesium-catalyzed polyamide 6 rods cast in different diametres, which are commonly used for gear manufacturing. Its toughness cannot be regarded as a constant feature due to the casting technology. The mechanical properties of the semi-finished products are sensitive to the manufactured dimension, e.g., cast diameter, which are investigated by the Charpy impact test and tensile impact test. It is generally accepted that the impact strength and tensile-impact strength correlate with the degree of crystallinity beside many other material’s feature. Crystallinity is evaluated by Differential Scanning Calorimetry. The aim of this study is to determine the relationship between toughness and crystallinity of the magnesium-catalyzed cast PA6 rods with different diameters. For the research cast rods between 40 and 300 mm diameter were selected in seven-dimensional steps. Based on the results, it was found that the toughness depends strongly on the diameter size. Furthermore, it is proved that the crystallinity explains 62.3% of the variation of the Charpy’s impact strengths, while the tensile impact method was not suitable to detect the difference between the test samples.

## 1. Introduction

The magnesium-catalyzed cast polyamide 6 (PA6) is a general use semi-crystalline engineering thermoplastic with some attractive physical, mechanical and tribological properties that provide a wide range of uses including industrial applications, e.g., gears among many automotive products. For gear application, the toughness is essential requirement. The small and medium scale production of cast polyamide gears is based on cutting technologies of semi-finished rods in a large variety of diameter range. It is essential to get to know the differences in toughness properties related to the diameter range. According to the published polymer literature, the properties of polyamide, i.e., toughness, are determined by the ratio of amide groups to the methylene group, the degree of crystallinity, the crystal morphology and the residual monomer [1,2]. It is critical to discover the characteristics and limitations of a cast PA6 that increase the possibilities in its practical applications. It is well known that based on the technical practice, the toughness and deformation ability of the cast PA6 strongly depends on the operating conditions and the manufacturing technology as well [3,4].

On the one hand, the moisture content and the operating temperature are vital from the point of view of the operating conditions. Polyamides are hygroscopic due to the amide groups, which can bond water. Thus, they absorb relatively a lot of moisture—from surrounding atmosphere during storage and use—compared to other thermoplastic polymers. The moisture is known to affect a range of polymer properties, in their mechanical properties such as elasticity, tensile strength, impact strength, as well as the performance of the semi-finished products [5,6,7]. In particular, the absorbed water in a polymer behaves like a plasticizer that reduces the entanglement and bonding between molecules, for this reason, increases their volume and mobility [8,9]. Furthermore, it also increases the impact strength, which is an indicator of toughness. At the same time, the polymer experiences an increase of elongation and a reduction of stiffness and strength [10,11].

Jia et al. have discussed that the impact of moisture on tensile properties of nylon also depends on temperature [8].

Several papers have been published on the effect of moisture and temperature, an increase in temperature or moisture can cause similar results in properties of polyamide and its composites, which is a significant reduction in the polymer strength and increase its ductility [8,12,13,14]. Qin et al. have investigated the effects of temperature on Charpy impact behavior. Their study has discussed that the energy absorption increased at a higher temperature, and reduced at a lower temperature that facilitated matrix cracking within the composite samples [15,16]. The impact strength has been reported against moisture and temperature by Garbuglio et al. The impact strength increases slowly from −120 °C to room temperature and a sharp increase occurs around 20 °C [17].

On the other hand, the degree of crystallinity and the crystal structure (spherulitic structure) play an essential role in the mechanical properties of polyamides and they are influenced by the manufacturing technology [18,19].

The degree of crystallinity is a crucial factor to be considered in the appraisal of the absorption capacity of polymers. The extent of moisture absorption depends on crystallinity. The crystalline phase does not absorb water. Some studies reported that water absorption is proportional to the amount of the amorphous phase [20,21]. It is complicated to separate the effects of the degree of crystallinity from the morphology. More pronounced spherulitic structure accompanies an increase in crystallinity. The impact strength, the toughness and fracture stress of polymers decrease with increasing crystallinity [22]. As the spherulitic structure of such plastics increases as a result of slow cooling from the melt below the melting point [10,23]. The degree of crystallinity of PA6 depends on the crystallization condition, and most importantly the synthetic protocol to produce them [19].

However, the relationship between the values of impact and tensile impact strength and degree of crystallinity has not yet been investigated at the different rod sizes of magnesium-catalyzed cast PA6 semi-finished products. Concerning the unique magnesium-catalyzed casting technology, the specific production features and method will be introduced in Section 2.2. in detail. The introduced literature results were obtained by testing of sodium-catalyzed cast polyamide, but industrial practice shows an increase in use of the magnesium-catalyzed version.

Consequently, the current paper focuses on the relationship between the toughness and the degree of crystallinity as a function of the manufactured dimension at the magnesium-catalyzed cast PA6 semi-finished products.

## 2. Materials and Methods

### 2.1. Material

Magnesium-catalyzed cast polyamide 6 (PA6) was obtained from Quattroplast Ltd., Budapest, Hungary, in the form of semi-finished products (rods), and its trade name is DOCAMID 6G-H. The general properties of that PA6 are listed in Table 1. This polyamide was produced by the casting process with anionic polymerization of caprolactam (C_6_H_11_NO), combining outstanding material properties (e.g., with vibration damping, high impact strength and abrasion resistance). The anionic polymerization of caprolactam was carried out using magnesium as a catalyst and N-acetyl-caprolactam as an initiator [24]. The polymerization occurs at temperatures around 160–165 °C without applied pressure.

Compared to the sodium-catalyzed system, this method allows for greater toughness, and impact resistance may vary widely depending on the casting technology settings [25].

### 2.2. Casting Technology

The technology for the production of the magnesium-catalyzed polyamide is based on a rapid polymerization of anhydrous caprolactam in the presence of the magnesium lactamate catalyst and activator. It is important to emphasize that it is a gravity casting technology (not centrifugal), thus the casting tools are at rest in the furnace during the polymerization process. The magnesium-catalyzed polyamide is partially chemically cross-linked and thus has individual physico-mechanical properties superior to commercially available polyamide-6 monomer compositions.

The production of magnesium-catalyzed polyamide can be divided into two main groups of operations, the preparatory operations and the polymerization process.

The first step in the preparatory operations is the melting of the caprolactam. Melting and temperature control are carried out at temperatures of 135–145 °C and provide a neutral atmosphere using nitrogen flow during the whole process. The next stage of preparatory operations is the dehydration of caprolactam, in which the molten caprolactam is divided into two equal parts and after that, they are pumped in a vacuum-degassing chamber. Half of the melt goes into the catalyst reactor and the other half into the activator reactor. The melt is de-watered and volatile by boiling at a temperature above 140 °C under vacuum.

The next step is the preparation of the catalyst and activator solutions. The solid catalyst containing magnesium caprolactam was melted airtight in an oven at a temperature of 145–150 °C and pumped to anhydrous caprolactam melt at a temperature above 130 °C. The mixture was tempered at the casting temperature and homogenized by vacuum degassing. In preparing the activator solution, an isocyanate compound was added to the anhydrous caprolactam, and the mixture was pumped into the activator solution reactor and homogenized by vacuum degassing at the casting temperature. Finally, the molds were prepared. The inner clean surface of the casting tools was provided with a mold release agent (silicone oil and silicone grease), the lid was placed on it and then it was heated in an air circulation furnace to the casting temperature, which was 160–165 °C depending on the size of the rods.

The polymerization of caprolactam began by mixing the catalyst and activator solution.

The two prepared solutions were filled into the prepared molds in a 1:1 volume ratio using a nitrogen overpressure of 0.15–0.25 bar in a boiling degassing and a pump. The forms were closed after filling. The polymerization was completed in 25–30 min, depending on the concentration of the catalyst system, the initial temperature of the polymerization and the presence of additives. The chemical process of polymerization was followed by a subsequent crystallization step. The two exothermic processes increased the temperature of the polymerization mixture by 5–50 °C, depending on the material thickness and the insulation conditions. The degree of temperature rise was determined by the mass, shape and thermal efficiency of the casting for a given kinetic catalyst system. The tempering and cooling rate after casting depends on the diameter of the finished product, in general after 25–30 min the tools can be covered (removal of residual caprolactam vapors). The finished cast was removed from the furnace with the tool after at least 30 min. The tools were emptied after 24 h, as there is a risk of subsequent deformation of hot-raised products (e.g., thin rods below Ø100 mm).

### 2.3. Testing Methods

The mechanical properties of PA6 were investigated by a Charpy impact test and tensile impact test. Charpy impact tests were carried out on INSTRON CEAST 9050 Impact Pendulum instrument according to the EN ISO 179 standard at ambient temperature. The span length was 62 mm. The energy of the hammer was 1 J, and the impact velocity was 3 m/s. The absorbed energy was registered during measuring, and the impact strength of notched specimen *α_c_* was calculated according to Equation (1):(1)$$\alpha_{c}=\frac{E_{C}}{h\cdot w}\cdot 10^{3},$$
where *h* is the thickness, *w* is the width of the specimen and *E_c_* is the energy absorbed by the specimen.

The tensile impact tests were executed on INSTRON CEAST 9050 Impact Pendulum testing machine according to the EN ISO 8256:2004 standard at ambient temperature. The energy of the hammer was 15 J, and the impact velocity was 4 m/s.

Crystallization and melting behavior of the polyamide 6 were characterised by a METTLER TOLEDO DSC 1 Star eSystem in a nitrogen atmosphere. Samples having a size of 5–20 mg were tested. The Differential Scanning Calorimetry (DSC) crucibles were applied with perforated lids to maintain atmospheric pressure during the measurement. Samples were first heated from −10 to 290 °C and then cooled down to −10 °C. The cooling and heating rates were 10 °C/min. The normalised crystallinity (*χ_c_*) of PA6 was determined by Equation (2):(2)$$\chi_{c}=\frac{\Delta H_{m}}{\Delta H_{100\%}}\times 100\;\left[\%\right],$$
where Δ*H_m_* is the enthalpy of fusion measured at the melting point and Δ*H*_100%_ is a reference value, which represents the enthalpy of fusion of totally crystalline polymer measured at the equilibrium melting point [26]. This reference of heat of melting for PA6 is 230 J/g [27,28].

Five parallel measurements of each sample were performed in all tests.

### 2.4. Sample Preparation

The tested rods were stored for one month after production in warehouse conditions, which promotes internal stress relaxation.

The samples were machined from 7 different diameter semi-finished rods, which were 40, 60, 90, 130, 170, 200 and 300 mm respectively. Due to structural differences coming from the casting production technology, the semi-finished rods were divided into skin and core zones. The 10 mm skin zone layer was removed. The upper and lower regions of the semi-finished rods were also removed. Furthermore, the rods were divided into three zones, as shown in Figure 1. Most samples were prepared from only the core zone with axial orientation by the milling process, except for those samples, which made for a second investigation of the 300 mm rod. These specimens were prepared from the skin and core zone as well. All specimens were manufactured at 23 °C and relative humidity of 50%.

The prepared specimens, after manufacturing, were conditioned in a desiccator for three weeks at 30% RH and 23 °C to keep their moisture content constant before measurement. Desiccator contained freshly activated silica gel, and it was sealed with grease to ensure saturation levels at all times. All specimens were arranged in a single layer; thus the surface was equally exposed to the air.

For the Charpy impact and tensile-impact tests, standard samples were made with a 4 mm × 10 mm cross section. The notch was ‘B’ type with 1 mm fillet radius, 2 mm depth and 45° angle. The notches were cut by an Instron Manual Notching Machine. The length of specimens was 80 mm. The Charpy specimen is single notched, however, the tensile-impact specimen is double notched. The standard allows multiple notch types for the test. Based on the preliminary experiments, the B-type notch of standard was decided because of the sensitivity of the crack propagation.

## 3. Results and Discussion

### 3.1. Charpy Impact and Tensile Impact Test

There are several types of polyamide 6 available on the market. Most of the studies in the literature relate to sodium-catalyzed cast and extruded polyamide 6. The values of their Charpy’s impact strength are under 10 kJ/m^2^ based on product catalogues for pure, notched sodium-catalyzed polyamide 6 samples according to the EN ISO 179-1/1eA standard [29]. The Charpy notched impact energy absorbed of the all samples at each investigated diameter with different zones (I, II and III) of the rods is shown in Figure 2. The average Charpy’s impact strength, which is expressed as mean values of 105 independent replicates, in the investigated size range was 15.9 kJ/m^2^ and its percent standard deviation was 10.5%. Thus, the magnesium-catalyzed polyamide 6 had a relatively higher Charpy’s impact strength than the sodium-catalyzed polyamide 6 had. Based on the Charpy’s test results (Figure 2.), it was concluded that the impact strength values were not permanent in the investigated rod diameter range. Moreover, they could be grouped into three categories, which was confirmed in Section 3.3. by the Dunnett’s test. To the first category, the rods with a diameter from 40 to 90 mm belonged here and their impact strength were between 16.2 and 18.5 kJ/m^2^. It was higher than the average. Second category, including the 130, 170 and 200 mm diameter rod, had a range of impact strength between 14.7 and 16 kJ/m^2^. The rod, which had a 300 mm diameter, fell into third category because its impact toughness was below the average, ranging between 12.5 and 15 kJ/m^2^. All data are mean values of at least five repetitions. It is indicated that the reduction of impact strength was able to reach 23%, it means 4 kJ/m^2^, if the rod with a 60 mm diameter was compared to a 300 mm diameter. From the point of view of the engineering application, the results suggest that the original size of semi-finished rods could influence the toughness behavior of parts.

There is another test that was good at evaluating the toughness of polymers, especially moulded materials and semi-finished products. It is the tensile impact test, which is appropriate to investigate the behaviour of specimens under high impact velocity. With this test, it is moreover possible to appraise the mechanical anisotropoc behaviour. The results of the tensile impact test in the same diameter range, like at the Charpy test, are presented in Figure 3. These results describe a dissimilar tendency as compared to the Charpy test results. While the effect of the sampling position within a rod was evident (except diameter 170/III), the differences among the diameters could not be demonstrated very well by the tensile impact test. There are more reasons for why it was not able to evaluate the toughness of cast PA6 with this kind of method. On the one hand, the standard requires a double notched specimen for the tensile impact test. Thus, the cross section of specimen is reduced as compared to the Charpy’s specimen. This difference in the cross section causes the specimen to become sensitive to mechanical stress. On the other hand, the type of stress is tension in the tensile impact test and the direction of stress is different, as well, to the arrangement of the specimen in comparison with the Charpy test.

For the Charpy test, the specimen was single notched, that means the remaining cross section is larger and when the pendulum impact against the samples in from of its notch that results in a bending stress at failure. Therefore, increased impact energy needs for break, thus, the differences among the diameters can be observed well.

The observed differences in toughness were analyzed by the Scanning Electron Microscope (SEM) images about fracture surfaces that can be viewed in Figure 4 and Figure 5. According to the SEM micrographs, it could be concluded that the number of fracture planes and crack initiations were less with higher deformation on the fracture surface of the specimen, which were made from 60 mm rod having the highest impact strength. The fracture surface had a lot of pattern respectively U- or V-shaped ramps in multiple occurrences that indicate ductile material behavior with plastic deformation (Figure 4.). The border of these fracture patterns is deformed wavy in some places due to the plastic deformation [30].

On the contrary, the specimen, which was made from 300 mm diameter rod with the lowest impact strength, indicates a different fracture surface that was smoother. It can be characterized with low deformation behavior and a large number of fracture planes (Figure 5) [31]. The other magnifications can be found in Appendix B.

Due to the significantly lower Charpy’s impact strength of the 300 mm rod, further detailed experiments were carried out where the samples were compared from the skin and core zone as well (Figure 1). Its results are documented in Figure 6.

According to the principle of manufacturing technology, semi-finished rods are made by quasi-adiabatic polymerization. In the range between 40 and 300 mm, the heat conditions differ at the casting technology set.

The reason for the deviation of the impact strength values originates from the casting technology. The materials are theoretically formed by adiabatic polymerization, which means that the rod is heated from an initial temperature through an exothermic process. The smaller the diameter of a rod, the less true is that the system is adiabatic. The warming up of a given diameter rod and the thermal equilibrium of system during the polymerization process increasingly depends on more external conditions. One of the most important factors is the actual diameter, e.g., the difference of a 50 mm diameter rod between the skin and core zone is less than for a 300 mm rod.

For the larger diameter of a rod, the core zone can heat up by an extra 40–50 °C as compared to the skin zone due to the poor thermal conductivity of the plastics. Ultimately, the thermal dynamics of the technology will determine the mechanical properties of the product, including the impact strength characteristics.

The results of detailed larger rod test have proven that the toughness of the skin and the core zone show significant dissimilarities, which is proved in detail in Section 3.3. by Welch’s *t*-test. The lowest and highest Charpy impact strength were measured at the samples machined from the 300 mm diameter skin zone II and core zone III, respectively. The change was 23% expressing 3 kJ/m^2^.

### 3.2. Differential Scanning Calorimetry

The previous impact tests established that dissimilarities in the toughness were found at semi-finished rods. There are more factors that can influence the toughness of polyamide. One of these factors is the crystallinity, which can be investigated by Differential Scanning Calorimetry (DSC).

DSC analysis was performed to monitor the changes in the degree of crystallinity of the studied magnesium-catalyzed cast polyamide 6, as followed in Figure 7. Although more papers have been published on the degree of crystallinity of sodium-catalyzed cast polyamide 6 by DSC. However, the degree of crystallinity of magnesium-catalyzed cast PA6 have not yet been investigated. The degree of crystallinity for sodium-catalyzed PA6 is about 40–50% based on the literature data [32,33]. The degrees of crystallinity of magnesium-catalyzed semi-finished PA6 rods were analysed and its range between 15% and 27% was found. The all crystallinity data represents the mean values of five replicates. The higher toughness compared to sodium-catalyzed version may refer to the lower crystallinity and higher portion of the amorphous part.

Based on the DSC results in Figure 7, it was concluded that the change in the degree of crystallinity affects the Charpy impact strength. Accordingly, the higher degree of crystallinity is able to cause the reduction of the toughness. Earlier studies have found the same [22,23]. This phenomenon can be definitely observed at the larger rods than 60 mm diameter. Furthermore, it was determined that the degree of crystallinity, from among the factors that influence the properties of polyamide, affected 62.3% of the variation of Charpy impact strength, which was proved in Section 3.4. by linear regression. Consequently, the other factor, i.e., the ratio of the amide and methylene group, the crystal morphology and the residual monomer etc. were responsible for 37.8% of the variation of Charpy impact toughness.

The detailed study of 300 mm diameter rod is reported in Figure 8, where S and C mean the skin and core zone, respectively. The degree of crystallinity was between 17% and 27% depending on the location of the sample taken along the cross-section. This 10% of variation in the degree of crystallinity may cause the difference in Charpy’s impact strength between the skin and the core zone. This data suggest that the Charpy’s impact strength decreased significantly with the increasing degree of crystallinity. It indicates that high crystallinity content was not necessarily advantageous for the impact strength in a 300 mm diameter or more region of magnesium-catalyzed semi-finished rods.

As already mentioned in the casting technology description (Section 2.2). The temperature adjustment range of the polymerization furnace could not bridge the heat fluctuations between the smallest and largest diameters. The set temperature of the furnace influenced much slower or less of the core zone of a large diameter rod. Thus, during the exothermic process, the temperature of the mixture might increase by up to 5–50 °C depending on the diameter and the insulation conditions. Therefore, the larger diameters took longer to polymerize, so that crystallization occurred in different ways. As it is known the degree of crystallinity depends on the conditions of the cooling method. Besides, for large-diameter blocks, there is a difference in crystallinity even in the skin and core due to the poor heat transfer.

The DSC curves can be found in the Appendix A.

### 3.3. Statistical Analysis

The presented statistical analyses below and in the next subsection were carried out by using version 25 of IBM SPSS. The detailed files of the statistical analysis are in the Appendix A.

First, the phenomena presented in Figure 2 was considered, which suggests that Charpy impact energy absorbed depends on the diameter. In general, to test whether a variable depends on a grouping factor, Analysis of Variance (ANOVA) was used. One of the assumptions of the application of ANOVA is homoscedasticity, which is the variance in groups should be the same. Homoscedasticity can be justified with a Levene’s test of homogeneity of variances. If heteroscedasticity occurs, the means of each group should be compered pairwise (it can be done in one step for example with a Dunnett’s T3 test). Levene’s test shows that at least for one diameter the variance of the measured data significantly differed from the others’ (F=3.040, p=0.009). In Table 2, the results obtained by the application of Dunnett’s test are summarized, the table contains the test values and an asterisk indicates if a value is significant at *p*-level 0.05 (since the matrix containing all the test values is antisymmetric, it is sufficient to present only the values in the upper right triangle of the matrix).

These results show that considering the Charpy’s impact, three significantly different groups could be formed with respect to the diameter. The first group contains the rods with low diameters, which are less than or equal to 90 mm, the second group stands of medium diameters (between 130 and 200 mm), while the only category in group 3 is 300 mm.

Next, it was examined that in the case of rods of diameter 300 mm whether the Charpy’s impact indeed significantly depends on the type of zone (core and skin), according to Figure 6 one might suggest this relation. To this end a Welch’s *t*-test was applied, since the variances of the measured data were unequal (F=13.041, p<0.001). The test’s result verified the conjecture (t=−20.076, p<0.001).

### 3.4. Regression Models

In this section the means of the single measured values belonging to the specific diameters were considered.

Table 3 presents the correlation coefficients of Charpy’s impact, diameter and crystallinity, we note that all the coefficients were significant at a *p*-level of 0.01.

As it is expected, the correlation coefficient of crystallinity and diameter turned out to be high because diameter includes more factors that influences Charpy’s impact and one of them is the crystallinity. Since these two variables are collinear, they cannot appear in a multivariate linear regression model at the same time where Charpy’s impact is the dependent variable, furthermore diameter and crystallinity are predictors. The crystallinity was chosen for predictor because it is a critical factor based on the literature. Thus, the following linear regression model was used (3):(3)$$C=a_{0}+a_{1}c,$$
where C denotes Charpy’s impact and c denotes crystallinity and $a_{0}$ and $a_{1}$ are coefficients to be determined using the method of least squares.

An analysis of variance was applied to check whether the model is relevant. Since F=16.397, p<0.001 it means that our model was applicable, which (in case of this simple model) coincides with the fact that the correlation coefficient of the Charpy’s impact and crystallinity was significant as well. Table 4 summarizes the coefficients of the model.

With the specific values Equation (4) turns into the following:(4)C=24.855−0.423c,

The measured data and the fitted function are presented in Figure 9. 

The goodness of fit of the model was *R*^2^ = 0.623, which means that crystallinity on its own explained 62.3% of the variance of Charpy’s impact. Note that single variate regression models with higher order terms of crystallinity was investigated as well, but no significant increase in *R*^2^ was experienced, therefore the discussion of such models were omitted.

## 4. Conclusions

After the investigation of the magnesium-catalyzed cast polyamide 6 semi-finished products in size of diameters ranging from 40 to 300 mm, the following conclusions were deduced:

The average Charpy’s impact strength of the magnesium-catalyzed polyamide 6 was 15.9 kJ/m^2^, which was higher than what the sodium-catalyzed version had.

Based on the Charpy test results, the investigated semi-finished rods had significant differences in their toughness and they could be grouped into three categories. To the first category, the rods with a diameter from 40 to 90 mm belonged here and their impact strengths were higher than the average ranging between 16 and 18.5 kJ/m^2^. The 130, 170 and 200 mm diameter rods were classified in the second category and they had a range of impact strength between 15 and 16 kJ/m^2^. The largest rod, which had a 300 mm diameter, fell into the third category where the impact toughness was lower than the average, it was between 12.5 and 15 kJ/m^2^. Moreover, in a detailed study of a 300 mm diameter rod, it was found that the toughness of its skin and core zone show significant dissimilarities.

Based on the DSC results, the crystallinity was only within 15–27% in the investigated diameter range. Furthermore, it was proved that the crystallinity was the major factor among the others that influenced the toughness of magnesium-catalyzed cast polyamide 6. The change in crystallinity explained 62.3% of the deviation of the Charpy impact strength. Additionally, the crystallinity of 300 mm rod was within 17–27% and this deviation might cause the difference in the Charpy’s impact strength between the skin and the core zone.

## Figures and Tables

**Figure 1 polymers-12-00293-f001:**
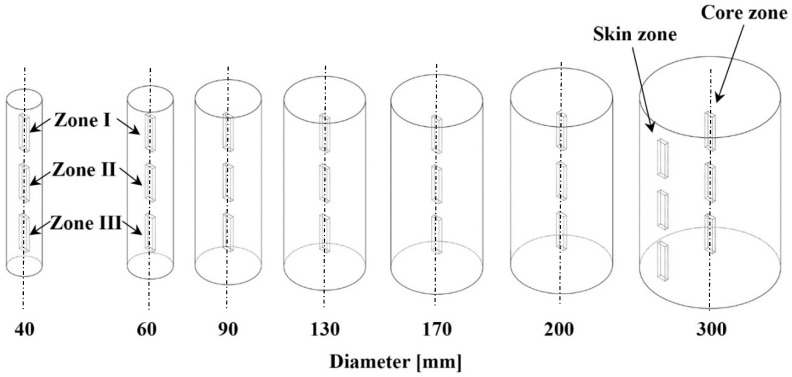
Scheme of the sample location.

**Figure 2 polymers-12-00293-f002:**
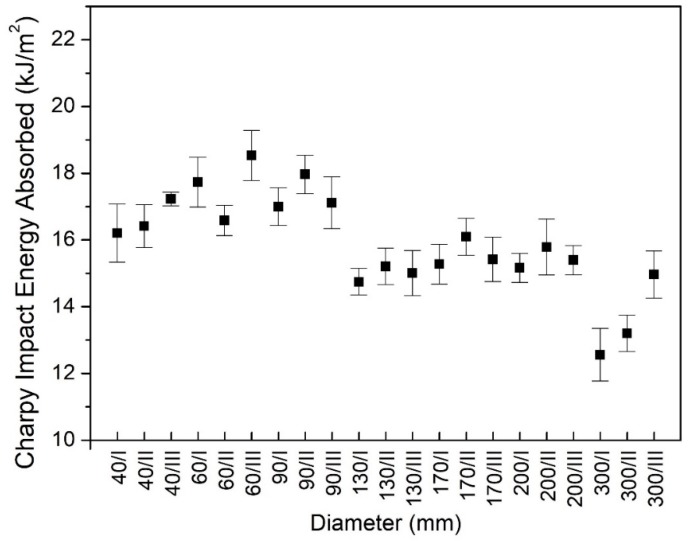
Charpy impact strength in the range from 40 to 300 mm.

**Figure 3 polymers-12-00293-f003:**
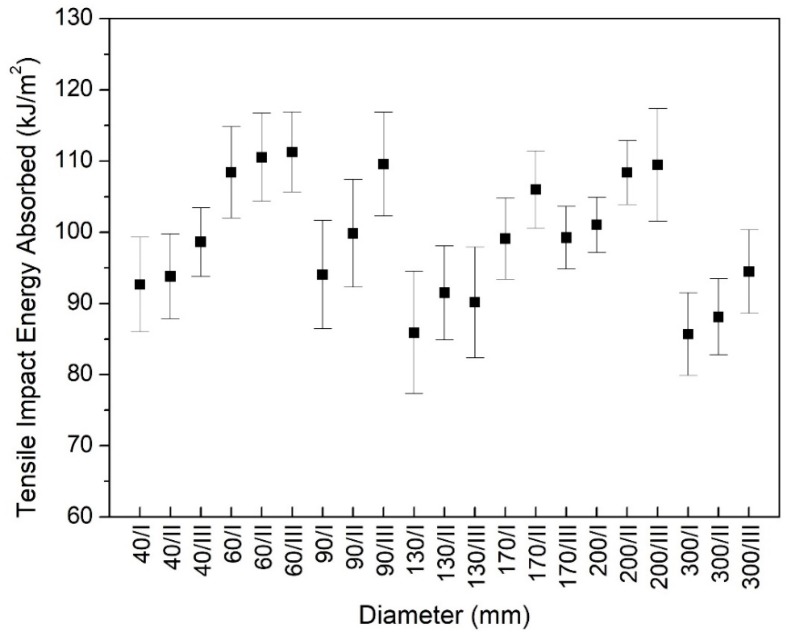
Tensile impact strength in the range from 40 to 300 mm.

**Figure 4 polymers-12-00293-f004:**
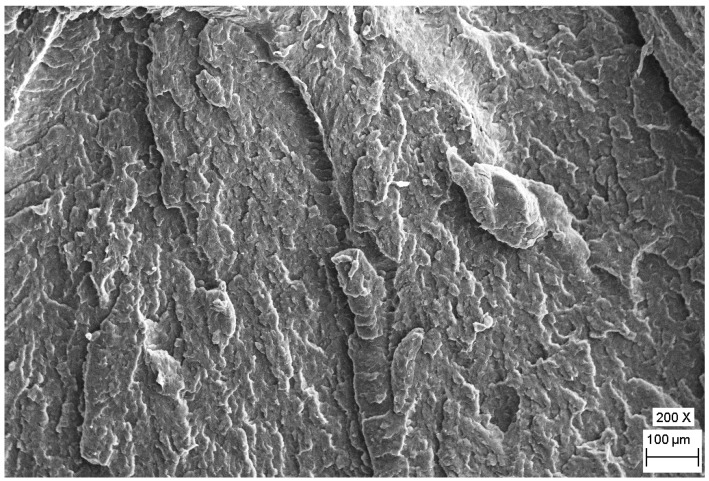
Scanning electron micrographs of the fracture surface of the Charpy specimens, the highest impact strength for the 60 mm rod at a magnification of 200 times.

**Figure 5 polymers-12-00293-f005:**
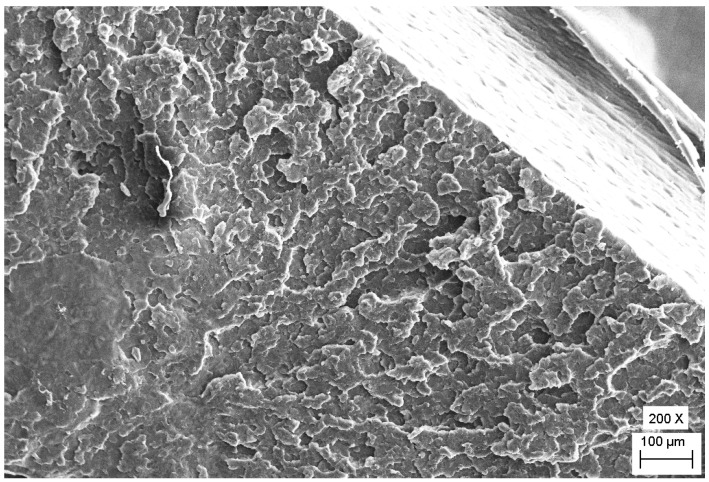
Scanning electron micrographs of the fracture surface of the Charpy specimens, the lowest impact strength for the 300 mm rod at magnification of 200 times.

**Figure 6 polymers-12-00293-f006:**
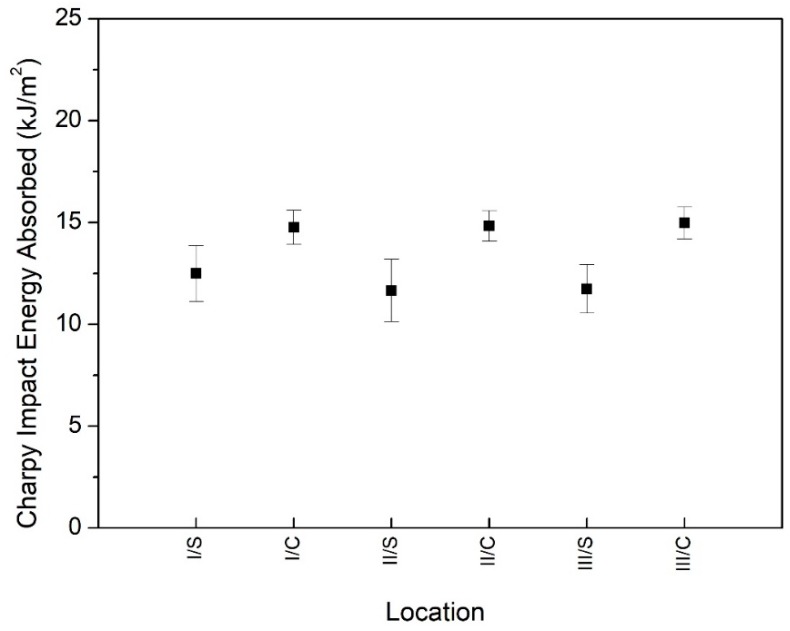
Detailed toughness test for a 300 mm diameter rod.

**Figure 7 polymers-12-00293-f007:**
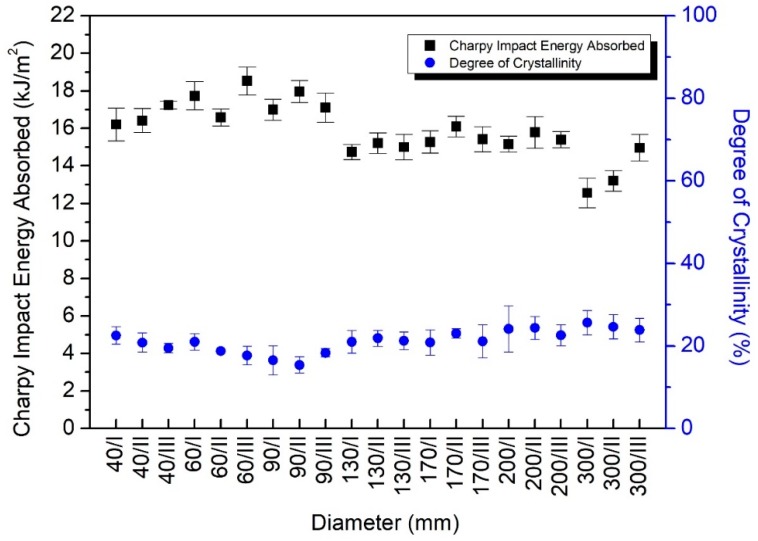
The relationship between the degree of crystallinity and Charpy impact strength in the case of magnesium-catalyzed semi-finished rods.

**Figure 8 polymers-12-00293-f008:**
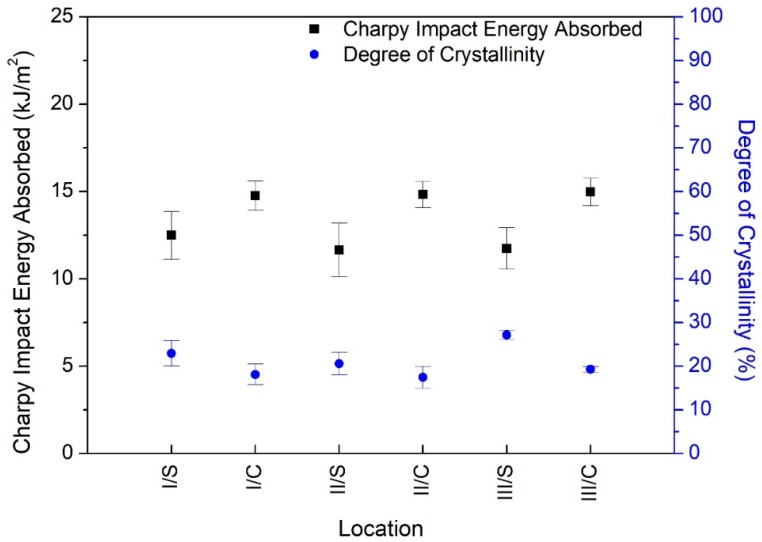
The relationship between Charpy impact strength and crystallinity in the 300 mm diameter of magnesium-catalyzed cast semi-finished rods.

**Figure 9 polymers-12-00293-f009:**
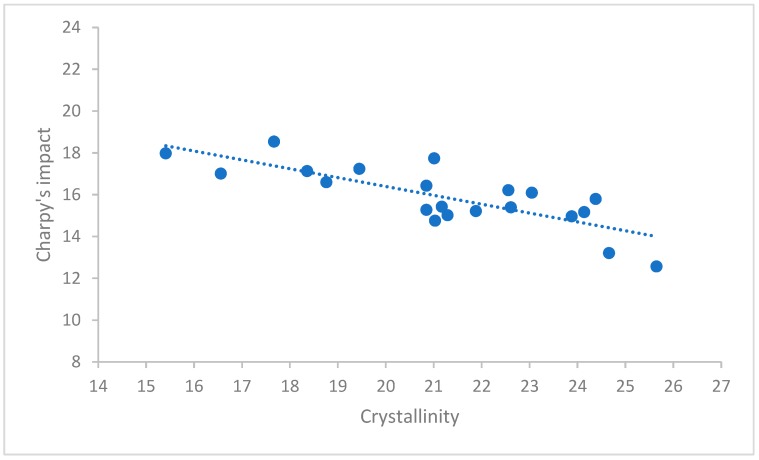
The measure data and fitted function.

**Table 1 polymers-12-00293-t001:** Characteristic properties of test material.

Property	DOCAMID 6G-H (PA6)
Density (g/cm^3^)	1.15
Yield stress (MPa)	85
Elasticity modulus (N/mm^2^)	3300
Thermal conductivity (W/mK)	0.38
Melting temperature (°C)	220
Glass transition temperature (°C)	40
Shore D hardness	81

**Table 2 polymers-12-00293-t002:** The results of Dunnett’s test.

Diameters	60	90	130	170	200	300
40	−0.999	−0.742	1.629 *	1.021 *	1.169 *	3.041 *
60	-	0.258	2.629 *	2.021 *	2.169 *	4.041 *
90	-	-	2.371 *	1.763 *	1.911 *	3.783 *
130	-	-	-	−0.608	−0.460	1.412 *
170	-	-	-	-	0.148	2.021 *
200	-	-	-	-	-	1.872 *

* The value is significant at the *p*-level 0.05.

**Table 3 polymers-12-00293-t003:** Correlation coefficient of Charpy’s impact strength, diameter and crystallinity.

Variables	Crystallinity	Diameter
Charpy’s impact	−0.789	−0.810
Crystallinity	-	0.726

**Table 4 polymers-12-00293-t004:** The coefficients of the linear regression model.

Model	Coefficient	*t*	*p*
Constant	24.855	15.405	<0.001
*c*	−0.423	−5.603	<0.001

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
