# Peer review of "The Effect of Crystallinity on the Toughness of Cast Polyamide 6 Rods with Different Diameters"

_polymers, 2020, doi:10.3390/polym12020293_

Round 1
Reviewer 1 Report
The content of this study is interesting, and the results are useful for the researchers in the related field.
However, there are some problems to be clarified. Therefore, I suggest a major revision of this paper.
More explanation about casting processing is needed and DSC curves should be provided as a supplementary file. According to Figure 6, crystallinity depends on the diameter of rods. Please explain the reason. Accroding to Figure 7, crystallinity depends on the skin and core. Please explain the reason
Author Response
Response to Reviewer 1 Comments
Point 1: The content of this study is interesting, and the results are useful for the researchers in the related field. However, there are some problems to be clarified. Therefore, I suggest a major revision of this paper. More explanation about casting processing is needed and DSC curves should be provided as a supplementary file.
Response 1: Thank you for pointing this out. I agree with these comments. Therefore, I have expanded the article with Section 2.2. about the casting processing. Furthermore, I have also uploaded the DSC curves as a supplementary file.
Point 2: According to Figure 6, crystallinity depends on the diameter of rods. Please explain the reason. According to Figure 7, crystallinity depends on the skin and core. Please explain the reason
Response 2: Thank you for these suggestions, too. I agree with them. I have modified the explanations of the crystallinity about the dependence on the diameter and the skin and core zone. The explanation can be found in the revised manuscript, Page 10, line 296.

Reviewer 2 Report
The article describes the effect of crystallinity on the toughness of PA 6 rods. The methods and the results of both Charpy and tensile impact tests for PA 6 rods are well described. However, the article has a few scientific components, so it is more like a good laboratory report than a scientific article.
In this paper, industrial PA 6 rods were used. Therefore, the article contains only common information about the conditions of obtaining PA 6 rods. The casting process conditions for obtaining PA 6 rod (temperature, pressure, casting speed, exposure) affect the final structure of product. It would be good to investigate the effect of PA 6 casting conditions on the formed structure and the properties of PA 6 rods with different diameters.
- What kind of supramolecular structure does PA 6 rods have? PA 6 supramolecular structure should be investigated by using additional methods (such as: X-ray, SEM after surface etching, etc.)
- In the SEM figure, the scale bar and the labels are small. The overall style of the Fig.4 should be improved.
- Page 6, line 188,the authors have written:“The fracture surface is more structured….”. What does “more structured” mean?
I believe that by adding more scientific components (such as: X-ray, SEM after surface etching, etc.) to the article, it could be published in the “Polymers”
Author Response
Response to Reviewer 2 Comments
Point 1: The article describes the effect of crystallinity on the toughness of PA 6 rods. The methods and the results of both Charpy and tensile impact tests for PA 6 rods are well described. However, the article has a few scientific components, so it is more like a good laboratory report than a scientific article.
In this paper, industrial PA 6 rods were used. Therefore, the article contains only common information about the conditions of obtaining PA 6 rods. The casting process conditions for obtaining PA 6 rod (temperature, pressure, casting speed, exposure) affect the final structure of product. It would be good to investigate the effect of PA 6 casting conditions on the formed structure and the properties of PA 6 rods with different diameters.
Response 1: Thank you for your suggestion. I agree with it. I revised my article and have expanded it with Section 2.2., which explains the casting process in detail.
Point 2: What kind of supramolecular structure does PA 6 rods have? PA 6 supramolecular structure should be investigated by using additional methods (such as: X-ray, SEM after surface etching, etc.)
Response 2: In our global research, the first goal was to clarify the mechanical (toughness) influence on the different diameter rods which are resulted by an industrial scale production technology. In this article, we could introduce the basic relationships between the impact strength and the crystallinity. However, chemical research and analyze the supramolecular structure belong to the further goals of this research. Thanks for your suggestion, which we take into consideration in the continuation of the research.
Point 3: In the SEM figure, the scale bar and the labels are small. The overall style of the Fig.4 should be improved.
Response 3: Thank you for the valuable remark. I have modified the SEM figure. I kept the pictures of magnification of 200 times in Fig. 4., and I moved the other magnifications to the Appendix.
Point 4: Page 6, line 188, the authors have written: “The fracture surface is more structured….”. What does “more structured” mean?
Response 4: Thank you for pointing this out. I wanted to emphasize many patterns in the fracture surface with “more structured” words. I have changed this sentence as follows: “The fracture surface has a lot of patterns respectively U- or V-shaped ramps in multiple occurrences that indicate a ductile material behavior with plastic deformation (Fig. 4 (a)).”

Round 2
Reviewer 1 Report
I believe the manuscript has been significantly
improved and now warrants publication in Polymers
Reviewer 2 Report
The main conclusion in the article is that the large semi-finished rods are not homogeneous, and therefore, they have different mechanical properties. However, this is obvious. The important and interesting issue is the understanding why this happens in more details and how could be avoided for this system. Without a structural investigation of PA6, this work can not be considered scientific article with a full value. I think that the obtained information could be interesting and valuable only to the manufacturer (Quattroplast Ltd.) and the consumer of polyamid 6, but in general, this article will not be of interest to a wide circle of researchers. Nevertheless, the article is well written, and I believe that in the future the authors will continue their work on this topic and will take into their consideration all the comments.